# Locally-curved geometry generates bending cracks in the African elephant skin

António F. Martins [1,2], Nigel C. Bennett[3], Sylvie Clavel[4], Herman Groenewald[5], Sean Hensman[6], Stefan Hoby[7], Antoine Joris[8], Paul R. Manger[9] & Michel C. Milinkovitch[1,2]

An intricate network of crevices adorns the skin surface of the African bush elephant, *Loxodonta africana*. These micrometre-wide channels enhance the effectiveness of thermal regulation (by water retention) as well as protection against parasites and intense solar radiation (by mud adherence). While the adaptive value of these structures is well established, their morphological characterisation and generative mechanism are unknown. Using microscopy, computed tomography and a custom physics-based lattice model, we show that African elephant skin channels are fractures of the animal brittle and desquamation-deficient skin outermost layer. We suggest that the progressive thickening of the hyperkeratinised *stratum corneum* causes its fracture due to local bending mechanical stress in the troughs of a lattice of skin millimetric elevations. The African elephant skin channels are therefore generated by thickening of a brittle material on a locally-curved substrate rather than by a canonical tensile cracking process caused by frustrated shrinkage.

[1] Laboratory of Artificial & Natural Evolution (LANE), Department of Genetics & Evolution, University of Geneva, Geneva 1211, Switzerland. [2] SIB Swiss Institute of Bioinformatics, Geneva 1211, Switzerland. [3] Mammal Research Institute, University of Pretoria, Hatfield 0028, South Africa. [4] Zoo African Safari, Plaisance du Touch 31830, France. [5] Department of Anatomy & Physiology, University of Pretoria, Hatfield 0028, South Africa. [6] Adventures with Elephants, Bela Bela D1000 LP, South Africa. [7] Zoo Basel, Basel 4054, Switzerland. [8] Réserve Africaine, Sigean 11130, France. [9] School of Anatomical Sciences, University of the Witwatersrand, Johannesburg, Braamfontein 2000, South Africa. Correspondence and requests for materials should be addressed to M.C.M. (email: Michel.Milinkovitch@unige.ch)

Because radiative and convective heat losses are limited by the generally warm and dry nature of its habitat[1], the African bush elephant (*Loxodonta africana*) thermo-regulates most effectively through evaporative cooling[2], which requires maintenance of its skin permeability by wetting because it lacks the sweat and sebum glands[3] that allow many other mammals to keep their keratinised outermost epidermal layer (the *stratum corneum*) moist and flexible. Hence, African elephants use regular bathing, spraying and mud-wallowing to hydrate their skin and promote direct and transepidermal evaporative loss to levels compatible with thermal balance requirements[1].

In addition to its characteristic wrinkling (Fig. 1a, b), the integument of the African elephant is deeply sculptured by an intricate network of micrometre-width interconnected crevices (Fig. 1c, d). This fine pattern of channels allows spreading (Supplementary Movies 1, 2) and retention of 5–10 times more water on the elephant skin than on a flat surface[4], impeding dehydration and improving thermal regulation over a longer period of time. In addition, this cracking pattern prevents shedding of applied mud[4], providing increased protection against

both parasites and intense solar radiation[5]. While the adaptive value of these skin channels is well established, their morphological characterisation and generative mechanism have remained unexplored. Using image analyses, microscopy and computed tomography (CT) on biological material, we show that African elephant skin channels are cracks that occur in the animal *stratum corneum*. In addition, our analyses suggest that this phenomenon is due to local bending stress caused by the African elephant's epidermis being simultaneously hyper-keratinised, desquamation-deficient and growing on a quasi-regular lattice of millimetric dermal elevations. Our custom physics-based lattice model confirms that the combination of these three parameters is sufficient to cause mechanical bending stress to accumulate in between the skin elevations during the progressive thickening of the *stratum corneum* until cracks are formed.

## Results

**Skin channels are fractures of the *stratum corneum*.** First, we establish that the patterning occurs at the level of the animal's *stratum corneum*. Indeed, the mechanical removal of this skin

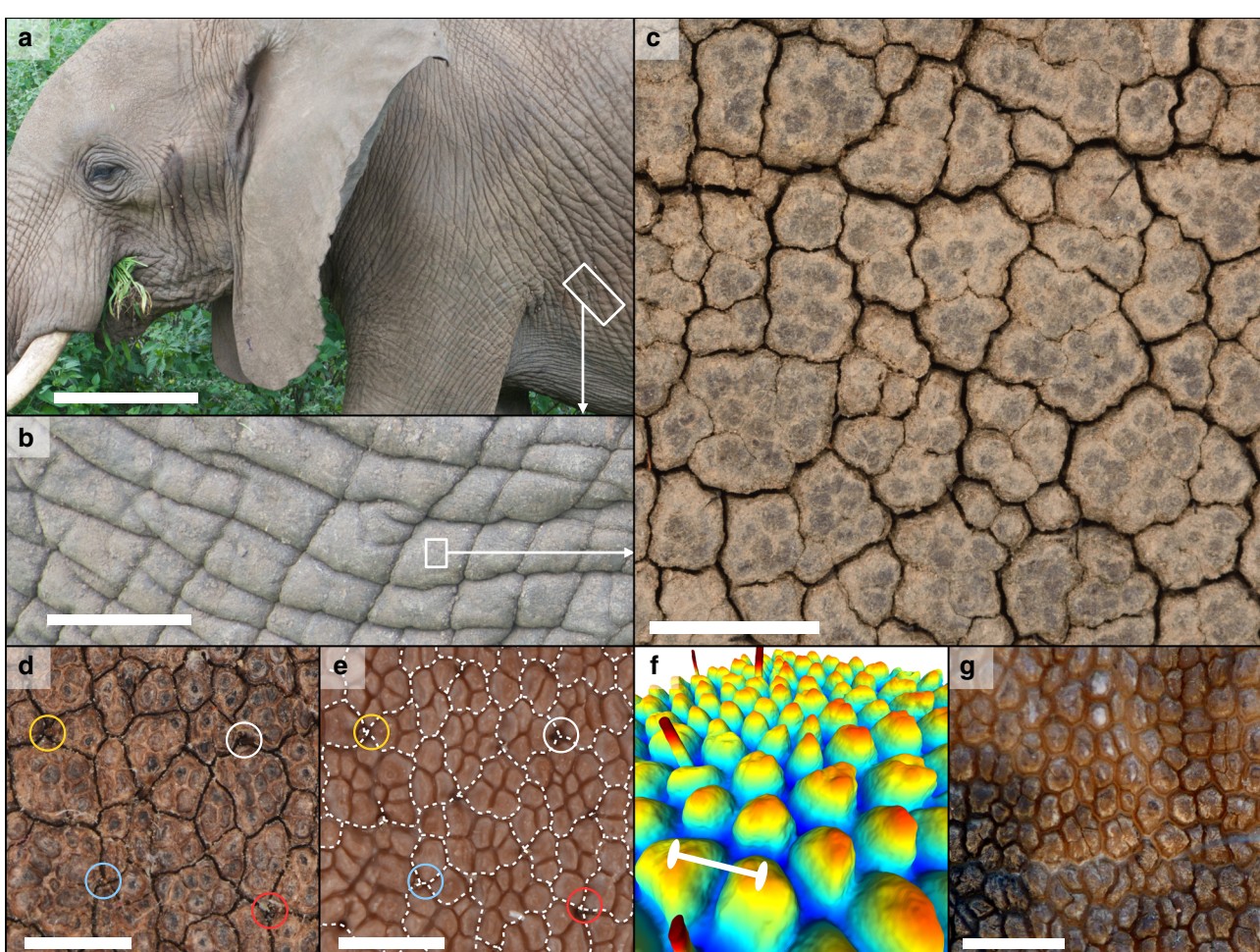

**Fig. 1** Morphology of the African bush elephant's (*Loxodonta africana*) skin. In addition to **a**, **b** centimetre-scale wrinkles and folds, African elephants exhibit **c** a network of narrow (~100 μm) channels (here, on the forehead of a juvenile animal). Previous research shows that these structures help the animal retain water and mud on its skin. **d**, **e** A patch of forehead skin before (**d**) and after (**e**) the mechanical extraction of the *stratum corneum*. In **e**, the narrow channels (white dotted lines) are no longer visible because they are confined to the extracted skin layer; note that the *stratum corneum* channels are confined in between papillae. The circled hair on both pictures serve as guides to the eye. **f** Mesh generated from the data of a micro-CT scan of a patch of skin without *stratum corneum* and colour-coded by height, showing the intricate three-dimensional surface geometry of the animal's papillae. **g** A patch of skin from the forehead of an Asian elephant. Even though the pattern of papillae is present, narrow channels are not observed. Scale bars: **a** ~50 cm; **b** ~7 cm; **c** 7.5 mm; **d** 2.5 mm; **e** 2.5 mm; **f** 1 mm; **g** 5 mm

layer results in a complete loss of the network of channels while it reveals a second underlying pattern composed of finger-like dermo-epidermal protrusions, hereafter referred as *papillae*, separated by lower-elevation *troughs* (Fig. 1d–f; Supplementary Movies 3 and 4). These structures form an intricately shaped substrate for the *stratum corneum*. Note that the channels in the *stratum corneum* appear to follow troughs in the underlying network of papillae (Fig. 1d, e). Although papillae are visible in Asian elephants (*Elephas maximus*; Fig. 1g), the pattern of channels is present uniquely in African elephants, suggesting that it is an adaptation to the more arid habitat of the latter[4].

Next, we investigate the nature of these crevices. Our light and electron microscopic analyses reveal that each channel in the African elephant skin is a sharp discontinuity in the *stratum corneum* (Fig. 2a, b). The keratin sheets of the latter display a clear alignment on both sides of the discontinuity, suggesting that, prior to the formation of the fissure, the *stratum corneum* was continuous in the trough, as observed in uncracked regions of the skin (Fig. 2c). Our proposition that channels in the African elephant skin correspond to physical cracking of the outermost highly keratinised part of the epidermis is strengthened by a diverse body of additional experimental observations that include: (i) the detachment of the *stratum corneum* into units delimited by channels when it is mechanically extracted from the rest of the skin (Supplementary Movies 3 and 4), (ii) the in vivo reorganisation of the network of channels following the regrowth of the *stratum corneum* after its ablation (Supplementary Figs. 1, 2), (iii) the heterogeneous trough width distribution visible in some samples (see Supplementary Text and Supplementary Fig. 3), and (iv) the presence of papillae but the absence of crevices on the skin of a newborn animal (see Supplementary Text and Supplementary Fig. 4). All these observations are compatible with our initial hypothesis, namely that the channels visible on the skin of the African bush elephant are physical cracks of the *stratum corneum*.

We then observe that African elephants exhibit orthokeratosis, i.e. a thickening of the *stratum corneum* (the thickness of which can easily exceed values of 300–400 μm; Fig. 2) without nuclei retention (Fig. 3), suggesting that there is an imbalance between its formation (at the *stratum basale*) and its shedding because of a deficient process of desquamation at the skin surface. Note that we borrow the term 'orthokeratosis' from the medical literature where it refers to a series of human pathological skin conditions, whereas the thickening of the *stratum corneum* in African elephants is the natural physiological phenotype. Additional insight is provided by our histology results (Fig. 3 and Supplementary Fig. 5), which suggest a lack of keratohyalin granules in the epidermis of the African elephant's skin. These characteristics (orthokeratosis and lack of keratohyalin granules) have been associated with a specific subtype of *ichthyosis vulgaris*[6], a skin disorder that is known to affect desquamation and cause dry, scaly skin in humans[7,8]. Although of heuristic nature, this parallel hints at the possibility that cracks occur in the skin of elephants because of the growth of a desquamation-deficient highly keratinised and brittle *stratum corneum* on a pattern of papillae that effectively acts as an intricately curved substrate (Fig. 1e, f).

**Locally-curved skin geometry generates bending cracks.** Concretely, we suggest that crack formation results from the following chain of events. As the *stratum corneum* forms, it acquires a naturally curved 'rest configuration' dictated by the geometry of the underlying dermis. Note that this configuration is likely permanent because the *stratum corneum* is composed of dead cells (corneocytes) that cannot rearrange topologically. Physically, this amounts to saying that, once formed, the *stratum corneum* behaves as a 'shell': a thin sheet of material that is naturally curved in its stress-free state. As more keratinous sheets are added at the *stratum basale*, the outermost sheets of the *stratum corneum* are effectively pushed outwards. As a result, and due to

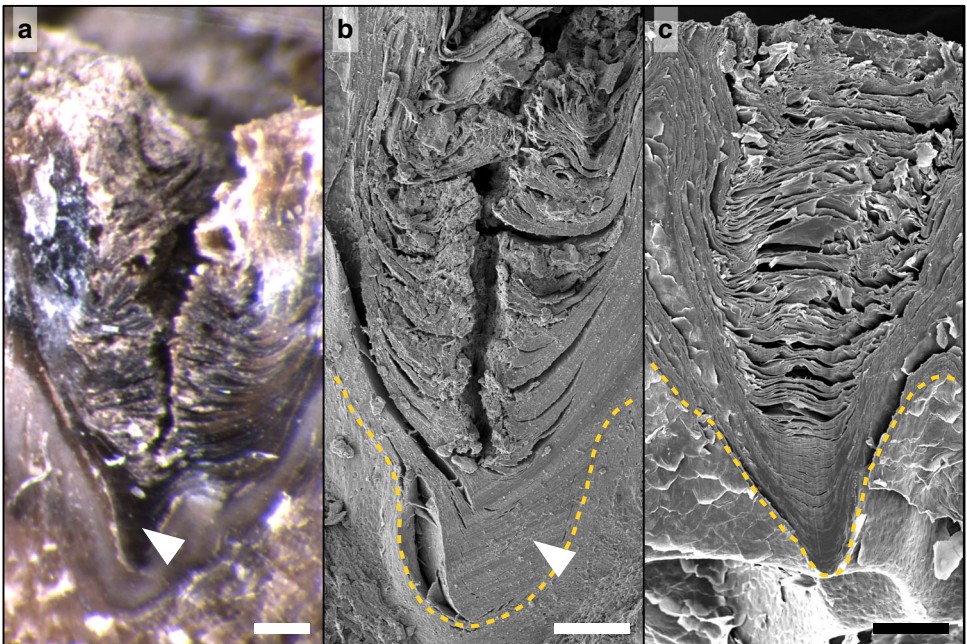

**Fig. 2** *Stratum corneum* morphology of cracked and uncracked samples. **a**, **b** Light (**a**) and electron (**b**) microscopy of sections cut perpendicular to a crack reveal a fracture that is characterised by an alignment of the *stratum corneum* keratin sheets on both sides of the fracture and a very small degree of recoil. Moreover, the innermost sheets of the *stratum corneum* remain intact (arrowheads), probably due to their higher hydration level. **c** Section cut through an uncracked sample of *stratum corneum*, showing a morphology identical to that of cracked samples apart from the lack of material discontinuities. Dashed lines in **b**, **c**: boundary between the *stratum corneum* and the viable epidermis. Scale bars: **a** 250 μm; **b** 150 μm; **c** 100 μm

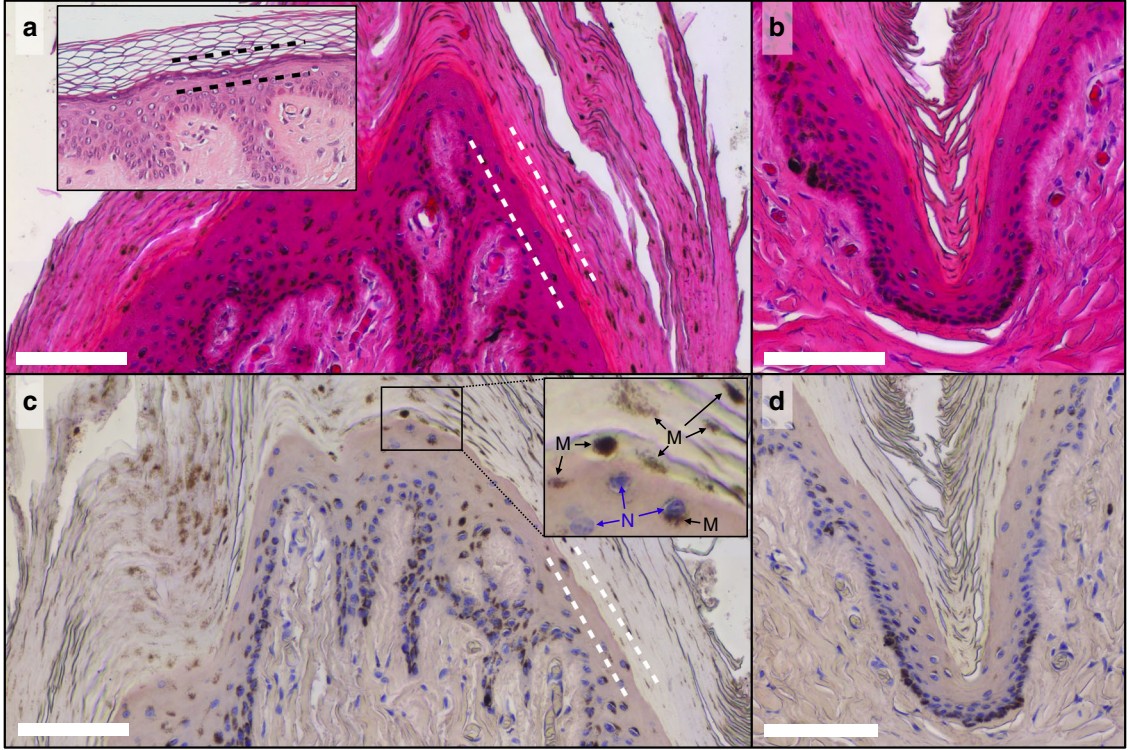

**Fig. 3** The epidermis of African elephants exhibits orthokeratosis and lacks keratohyalin granules. **a**, **b** Skin sections stained with haematoxylin and eosin from the top of a papillae (**a**) and from a trough (**b**). The lack of haematoxylin staining close to the transition from the viable epidermis to the *stratum corneum* (dashed lines) suggests that keratohyalin granules are not present in the animal's epidermis (inset, public domain image: similarly stained human skin section shows, between dashed lines, the strong haematoxylin staining due to the presence of keratohyalin granules). Orthokeratosis is equally perceivable: the *stratum corneum* is thick but otherwise normal looking. Similar morphological features are often observed in human patients with *ichthyosis vulgaris*. The inset in Fig. 3a is an adaptation of the image created by Kilbad available in the public domain, sourced from https://en.wikipedia.org/wiki/File: Normal_Epidermis_and_Dermis_with_Intradermal_Nevus_10×.JPG. **c**, **d** Same as **a**, **b**, but only haematoxylin staining was applied for better contrast. Again, the absence of keratohyalin granules is clearly visible. The inset of **c** indicates that dark spots in the *stratum corneum* are not viable nuclei (N) but clusters of melanin granules (M). Scale bars: 100 μm

the substrate's curvature, the outer sheets are forced to stretch, compress and bend, i.e. they become strained. These effects intensify as the growth of the *stratum corneum* progresses. In particular, one expects strong bending stresses to develop in the troughs. As partially dry *stratum corneum* fails in a brittle fashion[9], its outermost sheets are expected to reach the material's failure value, leading to the formation of cracks. Note that deficient desquamation prevents the *stratum corneum* to shed before it attains critical strain levels that commit it to material failure. The occurrence of this sequence of events across the skin of the African elephant eventually leads to the formation of a network of cracks.

To test this 'cracking-by-bending' hypothesis, we developed a custom quantitative physics-based model (see supplementary text) of a growing *stratum corneum* on an intricate skin geometry (i.e. exhibiting papillae) acquired from a micro-CT scan of a patch of African elephant skin (Fig. 1f, Supplementary Fig. 6). The objective of our simulations is to analyse the mechanical response (including fracture) of the *stratum corneum* to the growth of additional underlying layers and to connect it with the distinctive cracking pattern present on the skin of the African elephant. We model the outermost part of the *stratum corneum* as a shell (i.e. a thin sheet of material with a curved rest configuration) with a spring lattice and take into account the various factors and interactions that contribute to its deformation during growth (see Supplementary Text). These include its stretching and bending responses, its interaction with

the underlying skin (i.e. the 'substrate'; Supplementary Fig. 7), the possibility of self-contact and the formation of cracks by material failure. Contrary to finite-element approaches, our model handles the formation of discontinuities through a simple bond breaking mechanism, requires no geometry parametrisation and can easily be extended to include contact forces via, e.g. node–face interactions. Importantly, all the parameters of the model can be approximately expressed in terms of physically measurable properties and quantities, i.e. there are no fitting constants. While lattice methods have been extensively used in fields such as micromechanics[10], elasticity and plasticity analyses[11], fracture studies[12,13], membrane physics[14–16] and others[17], to the best of our knowledge this is the first time that they are applied to the study of shell structures. A full mathematical description of the model is provided in Supplementary Material file. More specifically, we describe in detail the lattice model in terms of energy contributions (spring, volumetric, bending, substrate and contact energies), the relation between the model and physical parameters, convergence to the steady state through damped Newtonian dynamics, the bond breaking procedure, the treatment of contact forces, how the mesh geometry and regularity is generated and the development of stress distribution and crack localisation and orientation. Note that we do not implement skin abrasion in the numerical model as, while definitely present (Supplementary Fig. 8), it seem to have little, if any, influence in the formation of the pattern.

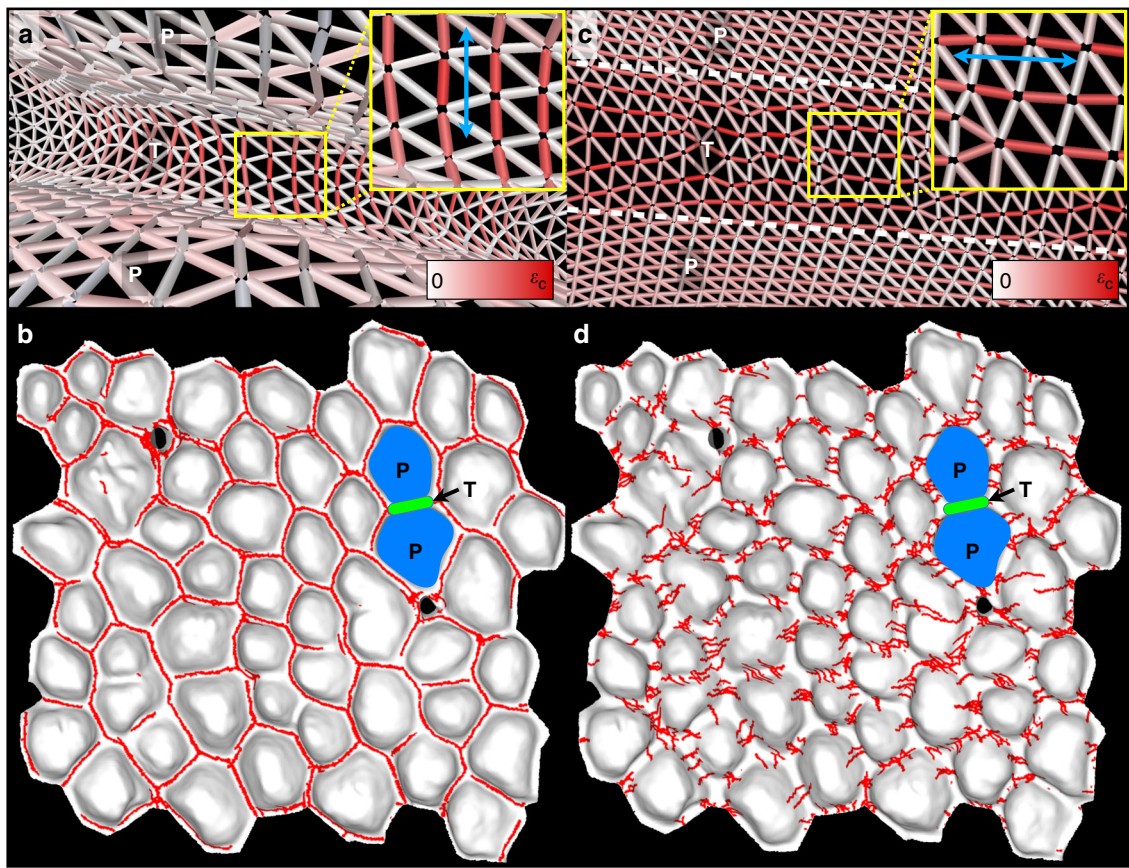

**Fig. 4** *Stratum corneum* cracking by growth versus shrinkage. **a**, **b** Under the model of growth and deficient desquamation, addition of keratinous sheets at the curved *stratum basale* makes the outer sheets experience strong bending stresses perpendicular (blue double arrow in inset of **a**) to the direction of troughs (T), causing cracks (red lines in **b**) to propagate in-between papillae (P). **c**, **d** Conversely, simulation of *stratum corneum* desiccation results in a cracking pattern (red lines in **d**) incompatible with those observed on elephants: simulated shrinkage makes the outer sheets experience stresses parallel (blue double arrow in inset of **c**) to the direction of troughs (T), causing cracks (red lines in **d**) to nucleate perpendicular to (rather than along) the trough directions, propagate over papillae (P) and not intersect at trough junctions. Mesh edges in **a**, **c** are coloured by the intensity of strain, from zero (white) to the critical strain ($\varepsilon_c$, saturated red) at which edges are broken. Results shown are for $\varepsilon_c = 0.3$ and $h/\langle d \rangle = 1/6$ ($\langle d \rangle$ = average trough spacing), at a stage when ~3% of the mesh edges are cracked

The results of our simulations provide quantitative confirmation for the hypothesis that the outer *stratum corneum*, when pressured by the growth of additional sheets underneath, develops strong bending strains and stresses perpendicular to the direction of the troughs (Fig. 4a; Supplementary Fig. 9). This is followed by the propagation of cracks along the troughs and the formation of a cracking network that is essentially confined to the regions in between papillae (Fig. 4b, Supplementary Movie 5, and Supplementary Figs. 10–11), consistent with what we observe on the animal (Fig. 1d, e).

The visual aspect of the cracking pattern on the skin of the African elephant (Fig. 1c; Supplementary Fig. 12a, b, d) bears a striking resemblance to canonical cracking systems in drying or cooling media such as mud and clay cracks, damaged asphalt (Supplementary Fig. 12c), polygonal patterned ground on the Earth's and Mars's polar landscapes, as well as columnar joints of the Giant's Causeway and Devil's Postpile, and their laboratory analogue, starch columns[18–25]. It is therefore tempting to hypothesise that shrinkage of the African elephant *stratum corneum* (through keratinisation and/or drying) generates cracks that propagate in between the skin *papillae*. Using the same lattice model as above, we tested this hypothesis by simulating desiccation of the *stratum corneum* on the same underlying substrate papillary geometry (Fig. 1f, Supplementary Fig. 6).

These computer simulations rejected the frustrated shrinkage hypothesis as they generated heterogeneous distributions of cracks that tend to propagate orthogonal to the direction of troughs (Fig. 4c, d), whereas cracks on the real skin occur near-exclusively along the troughs.

Using an in-house developed tool (see Methods and Supplementary Fig. 13), we then show that trough and crack junctions on African elephant skin exhibit distinct angular characteristics (Fig. 5a–d; Supplementary Figs. 14–16), the former's strong 'triple-120°' profile contrasting with the much wider angular dispersion of the latter. Our 'cracking by bending' model reproduces these differences: the simulated crack junctions, provided they are allowed to fully develop and mature (see Supplementary Text and Supplementary Figs. 17–19), exhibit angular characteristics that are quantitatively consistent with the ones observed for real skin cracks (Fig. 5c, d). Note that a qualitative understanding of this real and simulated crack junction wide angular dispersion can be obtained with the help of the simulated strain distribution (Supplementary Fig. 9): while a crack is intrinsically confined to a narrow space between two local skin elevations (i.e. a trough), it is given additional freedom (i.e. space) when it reaches a trough junction. The heterogeneous strain distribution in the through junction allows two propagating cracks to escape its centre and join with a roughly '90°–135°–135°' profile (Fig. 5a, b). Obviously, the

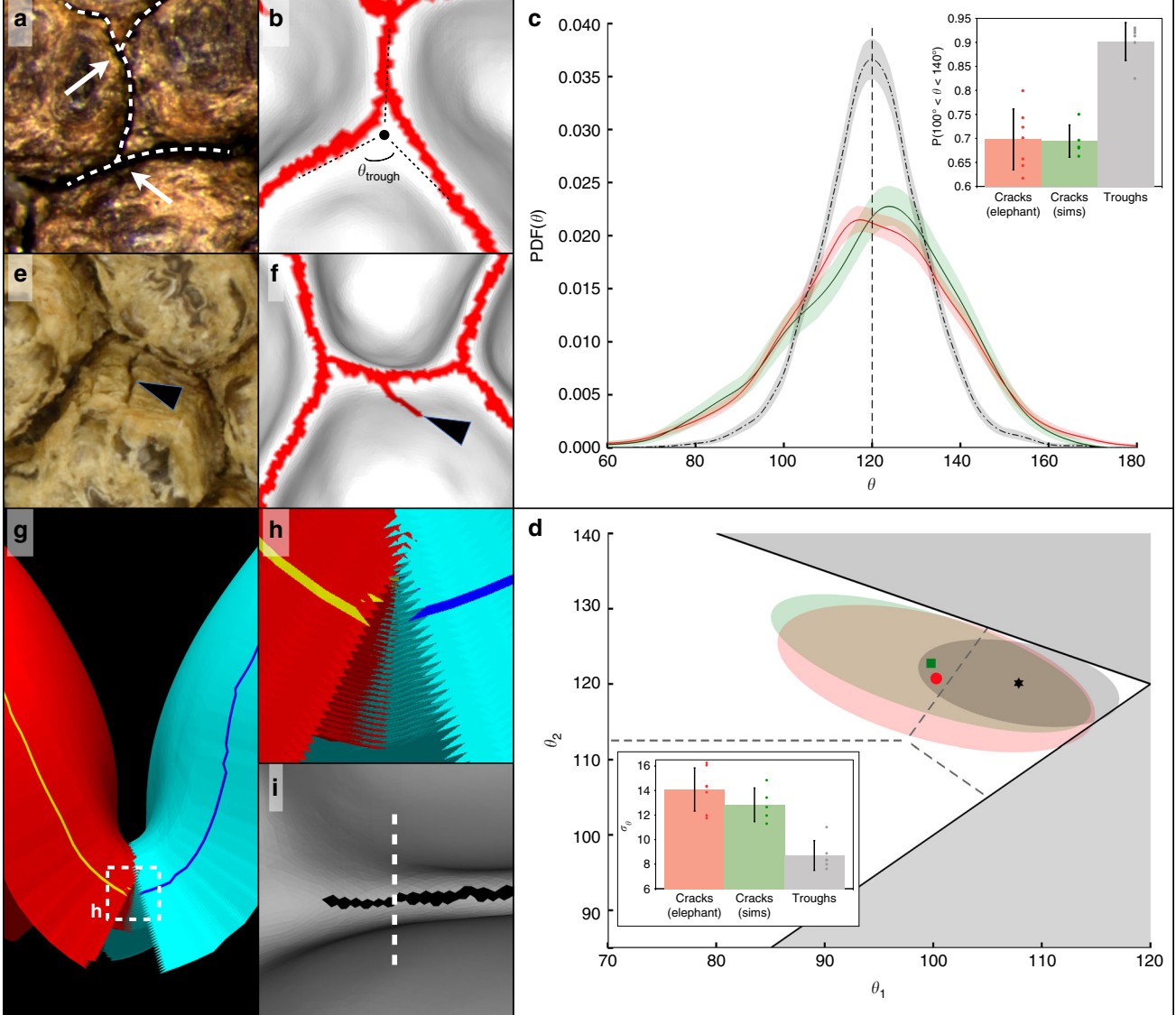

**Fig. 5** The physical model captures qualitative and quantitative features of the cracked skin pattern. **a**, **b** Crack junctions on the elephant skin (arrows in **a**) and in simulations (**b**) typically display an angular profile with a small (<90°) and two large (>135°) angles. Propagating cracks tend to avoid the centre of trough junctions (black dot in **b**) because of the heterogeneous strain distribution at the intersection (Supplementary Fig. 9). **c** There is a good fit between the probability density functions (PDFs) of the angular distribution of real skin (red line, $N = 2997$) and simulated (green line, $N = 966$) cracks, whereas they differ markedly from that of trough angles (black line, $N = 3645$). Shaded regions represent 95% confidence intervals. Inset: probability of finding an angle between 100° and 140°; mean (error bars = s.d.) among 7 PDFs of observed cracks (red, Supplementary Tables 3 and 4), 5 PDFs of simulated cracks (green, average of 193 angles per PDF) and 6 PDFs of observed troughs (grey, Supplementary Tables 2 and 4). **d** Angular scatter plot (shaded regions represent 50% confidence regions, see Methods) showing the 2D average and angular spreading of the first two angles of skin crack junctions (red point, $N = 999$), simulated crack junctions (green square, $N = 322$) and trough junctions (black star, $N = 1215$). The similarly oriented and dispersed angular profiles of skin and simulated crack junctions contrast with the 'triple-120°' trough profile. Inset: angular spreadings of crack and trough junctions; mean (error bars = s.d.) among 7 angular scatter plots of observed cracks (red, Supplementary Tables 3 and 4), 5 plots of simulated cracks (green, average of 64 vertices per plot) and 6 plots of observed troughs (grey, Supplementary Tables 2 and 4). **e**, **f** Rare cases of cracks propagating outside the troughs (arrowheads) in elephant's skin (**e**) and simulations (**f**). **g**, **h** Our physical model reproduces in silico the observed sheet alignment; yellow and dark blue lines represent the simulation surface, while the remaining sheets (red and light blue) form the thickness of the *stratum corneum*. The innermost uncracked sheets of *stratum corneum* are accounted for in the simulations (see supplementary text). **i** Top view of a crack (the dashed line indicates where the 'virtual section' (**g**, **h**) was cut)

narrower is the trough intersection, the more likely will be the crack junction to adopt a 'triple-120°' configuration.

Finally, our numerical model recapitulates the rare occurrence of side cracks (Fig. 5e, f), as well as the alignment of the *stratum corneum* sheets (Fig. 5g, h) and the lack of recoil between the two sides of the fracture (i.e. the width of the crack is of the order of the mesh spacing, see Fig. 5i), and suggests that the clumping of papillae (occurring in some body regions such as the forehead) is caused by increased thickness of the *stratum corneum* (see Supplementary Text and Supplementary Fig. 20). In all cases, we found the data from the real skin to be consistent with the proposed hypothesis of a cracking process generated by

bending stress, although we can not rule out the possible existence of spatial anisotropy of the *stratum corneum* (causing differences in its mechanical properties on papillae versus troughs) that would confine the cracks to the troughs.

## Discussion

In addition to standard patterning mechanisms such as positional information and reaction–diffusion[26–30], purely mechanical processes, such as folding[31,32], can also generate self-organised patterns in biological systems. On the other hand, while physical cracking patterns are commonplace in non-living materials, they are far less ubiquitous in biological systems. Cracking has been proposed as a patterning mechanism in the development of tree bark fissures[33,34], whereas the self-organised pattern of crocodile head scales has been suggested to form through an analogous process where skin bulges, generated by local cell proliferation, propagate in a tension field[35]. Here we show that the African elephant not only exhibits physiological and genuine physical cracking of its *stratum corneum* but that this process is also mechanistically distinct from the archetypal tensile cracking caused by frustrated shrinkage. Indeed, the elaborate network of micrometre-width cracks on the African elephant skin is likely caused by local bending stress due to the growth of a highly keratinised skin on a pattern of millimetric skin elevations.

Note that the process by which the papillae are patterned in the embryo is unknown. Further studies are also warranted to assess whether the lack of a cracking pattern on the skin of the Asian elephant is due to reduced orthokeratosis (in comparison to the African elephant), preventing the *stratum corneum* either from drying sufficiently to break in a brittle fashion or from reaching large enough thickness to generate material failure bending stresses in the troughs.

Our results also suggest that a parallel exists between the physiological characteristics of the skin of African bush elephants and that of humans affected by *ichthyosis vulgaris*. However, this potential equivalence needs to be validated by detailed molecular cell biology comparisons, including the confirmation in African elephants of two key markers of *ichthyosis vulgaris* in humans[6]: the lack of keratohyalin granules in the lower *stratum corneum* and a substantial decrease in the expression of profilaggrin, a precursor of the protein filaggrin involved in the aggregation of keratin filaments.

## Methods

**Animals**. Close-up photographs from a total of 16 specimens of African bush elephant (*Loxodonta Africana*) were acquired for image analysis and qualitative observations. These include data from living animals from zoos or reserves and fixed skin samples held by various institutions (Supplementary Table 1). In all cases, the pictures were taken with the consent of the animal's supervisor or the sample's curator. Additionally, we obtained museum formaldehyde-fixed skin samples from seven animals (IDs 3 and 11–16 in Supplementary Table 1) that were used, e.g. for histology, micro-CT scanning and statistical analysis of papillae. *Stratum corneum* regrowth was followed on a patch of skin located on the buttocks of animal 8, Nuanedi (see section 'Evidence for cracking as the patterning mechanism of the skin of the African elephant' in Supplementary Material). Animals' care was in accordance with international, Swiss and South-African institutional guidelines.

**Histology**. Formaldehyde-fixed tissues were dehydrated, embedded in paraffin, sectioned at 7 μm and stained with haematoxylin and eosin[36]. The stained sections were imaged using conventional bright-field microscopy, and when necessary, focus stacking was performed using Zerene Stacker's (Zerene Systems LLC 2017, Richland, Washington State, USA) depth map routine. To increase the resolution of the final result, several stacks were aligned and merged using a custom image registration script.

**Micro-CT sample preparation, imaging and mesh generation**. Micro-CT imaging was used as a tool to produce meshes with a realistic geometry for the numerical simulations. The *stratum corneum* was mechanically removed prior to micro-CT scanning in order to obtain the geometry of the quasi-regular lattice

of millimetric dermal elevations on which the *stratum corneum* grows. Skin samples were stained with 25% Lugol's iodine for ~48 h prior to imaging, as this procedure has been shown to produce good differential contrast with minimal tissue shrinkage[37]. Finally, samples were transferred to phosphate-buffered saline and imaged using a Quantum GX MicroCT Imaging System (PerkinElmer, Waltham, MA, USA) at 6 μm resolution. Raw data were analysed and segmented using Amira 6.2.0 (FEI, Hillsboro, OR, USA). The software was also used to produce a rough mesh, which served as the input for a custom mesh regularisation algorithm designed to smooth out the surface kinks and the irregular edge length distribution of the raw micro-CT mesh (see section 'Mesh geometry and regularity' in Supplementary Material). This regularised mesh output was then used in the numerical simulations.

**Angular scatter plots**. Because the two-dimensional (2D) networks of cracks only contain tri-junctions and the angles $\theta_1 \leq \theta_2 \leq \theta_3$ of a junction sum up to 360°, each junction can be represented by a 2D point ($\theta_1, \theta_2$). Such 'angular scatter plots' avoid the loss of information due to binning in density profiles. This methods also allows easy representation of the average and confidence regions of the angles $\theta_1$ and $\theta_2$ for a population of vertices. The angular spreading is computed as $\sigma_\theta = (A_{1\sigma}/\pi)^{1/2}$, where $A_{1\sigma}$ is the area of the one standard deviation (~68% confidence level) error ellipse. The allowed angular space is partitioned into three regions (delimited by dashed lines in the plots) obtained by calculating the Voronoi diagram generated by three archetypal junctions: 'triple-120°', '90°–135°–135°', and '90°–90°–180°' ('T-junction'). See Supplementary Materials for additional details.

**Numerical model**. Our numerical analyses rest on the assumptions that (i) the *stratum corneum* can be modelled effectively as an homogeneous material and (ii) little or no *stratum corneum* desquamation takes place on the skin of the African elephant, as suggested by the spectacular thickening of the animal's *stratum corneum* and the histological similarities between skin sections of African elephant and human patients with *ichthyosis vulgaris*. Given the animal's intricate skin geometry (i.e. the presence of papillae), this implies that the outermost part of the skin will be subjected to stress due to the growth of the underlying tissue.

A number of numerical methods have been proposed to simulate shells[38–41]. These are almost universally based on some variation of the finite element method. However, the inclusion of material discontinuities (cracks), the complex geometry of the animal's skin (i.e. the presence of papillae with no straightforward parametrisation) and the possibility of self-contact make this class of algorithms ill-suited to our study. To overcome these difficulties, we have developed a custom lattice spring model that takes into account the stretching and bending responses of the shell, the effects of a substrate with arbitrary geometry and the self-contact that can occur between different parts of the mesh. A detailed mathematical description of the model is provided in Supplementary Material file.

## Data availability

Data supporting the findings of this manuscript are available from the corresponding author upon reasonable request. Large files with simulation 3D geometries are available from the corresponding author on reasonable request.

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

## Acknowledgements

We thank Athanasia Tzika, Ingrid Burgelin, Liana Manukyan, Adrien Debry, Florent Montange and Valérie Haechler for technical assistance with histology, image acquisition and analyses, as well as Isabelle and Albert Haechler for wetting experiments. This work was supported by grants to M.C.M. from the University of Geneva (Switzerland), the Swiss National Science Foundation (grants 31003A_140785 and SINERGIA CRSII3_132430), the SystemsX.ch initiative (project EpiPhysX) and the Human Frontier Science Program (HFSP RGP0019/2017).

## Author contributions

M.C.M. conceived and supervised the study. A.F.M developed the angle marking toolkit and performed laboratory experiments, statistical and theoretical analyses and numerical simulations. M.C.M., A.F.M., S. Hensman, S. Hoby and S.C. photographed live animals. S. Hoby, A.J., N.C.B., P.R.M. and H.G. provided skin samples. S. Hensman performed the *stratum corneum* regrowth and wetting experiments. A.F.M. and M.C.M. analysed the data. M.C.M. and A.F.M. wrote the manuscript and produced the figures. All authors approved the final version of the manuscript.

## Additional information

**Competing interests:** The authors declare no competing interests.

