## [Peer Review File · Nature Communications]

Reviewers' comments:

Reviewer #1 (Remarks to the Author):

This paper describes the pattern formation of crevices in elephant skin. At first they present morphological and histological characteristics of the crevice shape. Then they argue the functional importance of the crevices, and provide a physical model of crack formation. They provide two hypotheses - shrinkage of the surface and growth of the inside, and showed that latter mechanism is probable because of the directionality of the cracks. They also undertook morphometry to support their model.

Although I agree the phenomenon itself is interesting, I think this manuscript is too premature to propose a mechanism of pattern formation. The mechanism itself is not very new, the key biological information is missing and model analysis is not enough. The authors should accumulate more data experimentally and add in-depth analysis theoretically before resubmission to better journal.

1. Biological aspect

Utilization of breakdown phenomena as a pattern formation mechanism is not very novel. About 80 years ago there is a severe controversy between Japanese biologists and physicists on whether giraffe skin pigment pattern can be explained as a fracture. At that stage there are not enough biological evidence of animal coat marking pattern formation. Later several biologists start to insist that the pattern can be reproduced by Turing model, which is fundamentally different from original physical fracture model (Kondo & Asal, 1995).

In addition, observation of pattern formation dynamics is necessary to understand the mechanism. The critical information to settle the controversy in the above example is the observation of pattern dynamics. They should provide some experimental data on when these crevices are formed during development. If they are seasonal then we should reconsider surface drying model. If the pattern corresponds to growth curve of the individual, then it should provide strong support for growth model. Sequence of crack pattern formation or scaling of crevice size is also very interesting.

If the pattern is due to the mechanical fracture formation, effect of movement should be observed. Crevice pattern in skins in axillary region or cubital region should have some anisotropy.

2. Theoretical aspect

From theoretical point of view, the paper is less interesting because it deals with crack formation of preexisting meshwork of valley and not the initial phase of valley itself. The phenomenon is later modification of preexisting pattern and much easier problem from pattern formation point of view. Comparison of two mechanisms should be understood by more analytical way. We can consider a elastic body whose shape is single valley (half pipe) and obtain tension at the surface by these two models analytically with given boundary condition.

Kondo, S., & Asal, R. (1995). A reaction-diffusion wave on the skin of the marine angelfish *Pomacanthus*. *Nature*, 376(6543), 765–768.

Reviewer #2 (Remarks to the Author):

The paper by Martins et al is an elegant and exceptional paper, which it would do credit to Nature Communications to publish. It is highly detailed and innovative, and the supplemental model is, itself, more detailed, accurate and rigorously tested than many fracture models that I have seen

published – it would make a quite respectable publication in its own rights.

I would recommend publication of this, without any need for further modifications. In a couple places, I found the language a little hard to follow (particularly given my physical sciences background), but this was usually something that could be understood by a careful second reading. Nonetheless, the authors may wish to better define what they mean by trough on its first use in the second paragraph of the results, as I first understood it to mean the trough in a fold on the skin, rather than between the smaller features. Second, although it is defined in the supplemental info, it would be helpful to explain the way angles are used and defined in Fig. 5 in the figure caption (i.e. that $\theta_1 > \theta_2 > \theta_3$, etc.). Finally, I'm afraid that I don't fully understand what is being presented in Fig. 4 a/c, and don't see what features in those correspond to the cracks, or what the dashed lines by the P's mean. All these are exceptionally minor details.

As a caveat, my background is in the physical sciences, in fracture mechanics and material responses to strains. I cannot comment on the biological rigour of this paper in a professional way. I would strongly recommend that someone from a more relevant background be taken on as a reviewer as well.

Reviewer #3 (Remarks to the Author):

In this manuscript the authors describe the morphological and physical structure of the african elephant's epidermis and a specific cracking behavior of it's outermost layer the stratum corneum that is thought to allow retention of mud and water. The authors have developed a potentially very interesting computational modeling tool that recapitulates the structure and cracking behavior of the stratum corneum and suggest that the specific cracking pattern is a result of orthokeratosis and reduced desquamation in combination with the specific topology of the underlying dermis resulting in local bending stress.

The description of the biophysical process as well as the model are of potentially high interest for epidermal biology not only for elephants but even for human diseased conditions. However, the causality of the biophysical features like e.g. topology of the dermal papillae, hyperkeratosis, desquamation and the cracking behavior is not completely evident, even though highly suggestive. Whereas the outcome of the computational modeling clearly supports the authors hypothesis, it is not always clear whether the assumptions match the biology. Thus, testing of the model based on another maybe contrary biological basis like e.g. Asian elephant skin would more strongly support the hypotheses stated here.

The authors suggest that crack formation is a result of dermal curvature, orthokeratosis due to deficient desquamation and bending stress. If one of those features is really causal, the prediction is that Asian elephant would lack any of those features.

The similarities between elephant and ichthyosis patient skin that have been mentioned do suggest a cracking stimulus that is not solely based on dermal topology but e.g. the moist state of the SC as discussed in the manuscript. The authors state that based on their modeling, cracks in the elephant skin cannot be explained by desiccation. Not being an expert for computational modeling nor physic it is difficult to review whether the parameters that were changed would reproduce natural phenomena as seen in e.g. dry skin phenotypes. Thus a model based on a CT from an Asian elephant would help showing that an altered topology or altered proliferation/desquamation behavior or altered corneodesmosomal adhesion explains the lack of cracks.

This questions have in part being addressed in the supplementary section by discussing differences of forehead skin or young animals. However, those questions should be addressed at a more prominent part of the paper as they address the biological nature of the elephant skin phenotype. Moreover, a comparison with Asian elephants would show better that the model could be used to

unravel variables that determine structural features of the SC or vice versa, to find causes for structural abnormalities as found in human patients.

Therefore, even though outside the scope of this manuscript, an analysis of human ichthyosis patients might reveal cracks being specific for desiccation, like "cracks that tend to propagate orthogonal to the direction of throughs" as shown in Figure 4. So could an analysis of SC crack behavior in human patients be used as diagnostic tool to find the origin of a barrier defect? If understood correctly the model does allow to discriminate cracking behaviors of e.g. differentially hydrated SC as suggested for different layers of the SC.

This study could be very helpful in understanding overall stratum corneum formation and function, if the different aspects of its physics, that have actually been nicely discussed the supplementary section, would be addressed more experimentally to aid the biological significance of the model.

Minor points:

It is not clear why the authors state that African elephants show signs of orthokeratosis, being an abnormal epidermal phenotype. If there is no other elephants epidermis this is compared to, it is not clear why this is not normal african elephant epidermis with a thick stratum corneum. This may indicate an altered balance of proliferation and shedding compared to human epidermis resembling a diseased human but not elephant phenotype. Thus the wording of this paragraph should be revised.

Ichthyosis vulgaris patients are characterized e.g. by reduced fillagrin levels resulting in a brittle SC. Are fillagrin levels reduced in African elephants or is fillagrin at all expressed?

Reviewer #4 (Remarks to the Author):

In this elegant and well written study, Martins et al. present a physics lattice model to demonstrate that micron-size channels in the skin of the African bush elephant are fractures of the stratum corneum caused by local bending mechanical stresses in troughs. This finding is intriguing because, after the corresponding author's earlier study on crocodilian head skin, this report provides another example—and the first to my knowledge in mammalian epidermis—of a mechanically encoded morphology based on cracking, rather than one based on purely genetically encoded parameters (though, of course, genetic parameters may still be in part at play).

This study, with quantitative imaging of a non-traditional animal model and a rigorous physical model, will be of interest to those working in a variety of fields, including the general readership of Nature Communications. The manuscript would be strengthened by consideration of following points:

Major comments:

- The link between African elephant skin and human ichthyosis vulgaris is weak and hand-waving at best and should be scaled back. The authors should consider other ways to motivate their hypotheses.
- More so than the images of cracked and uncracked elephant skin, the real contribution of the manuscript seems to be in the development and implementation of the physical model. I encourage the authors to include more information about the model in the main text. Currently discussion of the model encompasses just over one page (of five) of the main text. In the manuscript's current form, the supplementary materials about numerical model could stand on their own as a full article in a more specialized journal. For publication in Nature Communications, I would like the authors to strike a better balance that allows more of their contribution (the model) to appear in the main text, to counter some of the more hand-waving arguments about comparisons to, e.g., human ichthyosis vulgaris and Giant's Causeway. The final two paragraphs of

the Methods section are most intriguing to me, and I would like to see those ideas fleshed out more in the main text. Along those lines I would encourage some more of the supplementary panels to be moved into parts of the main figures.

- I am not convinced that the African elephant's epidermis lacks keratohyalin granules based on the data presented. Can the authors provide higher-resolution electron microscopy images and/or clearer stainings to show absence of these granules?

- I am not convinced that the African elephant's stratum corneum exhibits orthokeratosis, as I see small thin spots in Figure 3a,c that could be nuclei. Can the authors verify in another way that the stratum corneum is indeed anuclear?

Minor comments:

- Figure 4a,c needs a color bar showing what the different shades of red mean (as provided in Figure S9).

- Can the authors explain why the elephant and simulation data in Figure 5c peak on opposite sides of 120° ?

Orthokeratosis and Locally-Curved Geometry Generate Bending Cracks in The African Elephant Skin (ms NCOMMS-18-11472)

Answers to Reviewers (May 25, 2018.)

Rev#1

This paper describes the pattern formation of crevices in elephant skin. At first they present morphological and histological characteristics of the crevice shape. Then they argue the functional importance of the crevices, and provide a physical model of crack formation. They provide two hypotheses - shrinkage of the surface and growth of the inside, and showed that latter mechanism is probable because of the directionality of the cracks. They also undertook morphometry to support their model.

Although I agree the phenomenon itself is interesting, I think this manuscript is too premature to propose a mechanism of pattern formation. The mechanism itself is not very new, the key biological information is missing and model analysis is not enough. The authors should accumulate more data experimentally and add in-depth analysis theoretically before resubmission to better journal.

Answer: We are sorry to disagree with the reviewer. In fact, cracking as a patterning mechanism in living organism, is quite new. We more specifically answer the criticisms of the reviewer below. Note also that the journal *Nature Communications* is definitely good enough for us.

1. Biological aspect

Utilization of breakdown phenomena as a pattern formation mechanism is not very novel. About 80 years ago there is a severe controversy between Japanese biologists and physicists on whether giraffe skin pigment pattern can be explained as a fracture. At that stage there are not enough biological evidence of animal coat marking pattern formation. Later several biologists start to insist that the pattern can be reproduced by Turing model, which is fundamentally different from original physical fracture model (Kondo & Asai, 1995).

Answer: We appreciate the historical perspective outlined by the reviewer regarding the possibility of cracking being involved in giraffe skin colour patterning (and then, the subsequent demonstration that reaction-diffusion-like processes are much more likely to be responsible of the observed pattern). But this remark from the reviewer is actually strengthening our case: contrary to the case of the giraffe colour pattern, we do demonstrate experimentally that the intricate network of narrow channels on the skin of African elephants does correspond to fractures in the *stratum corneum*. Hence, the crevices are generate by material cracking; we establish it as a fact, not as an hypothesis. This observation is entirely new. On the other hand, the reviewer is correct that our mechanistic explanation for the generative mechanism of these cracks (local mechanical bending stress in the troughs of the lattice of skin millimetric elevations) is an hypothesis, and it is presented as such throughout the main manuscript and supplementary material. Note that we now added in the discussion paragraph a sentence about other more common patterning mechanisms (such as Turing systems) together with some additional references, including Kondo & Asai, 1995.

In addition, observation of pattern formation dynamics is necessary to understand the mechanism. The critical information to settle the controversy in the above example is the observation of pattern dynamics. They should provide some experimental data on when these crevices are formed during development. If they are seasonal then we should reconsider surface drying model. If the pattern corresponds to growth curve of the individual, then it should provide strong support for growth model. Sequence of crack pattern formation or scaling of crevice size is also very interesting. If the pattern is due to the mechanical fracture formation, effect of movement should be observed. Crevice pattern in skins in axillary region or cubital region should have some anisotropy.

Answer: These remarks are well taken but these issues are addressed throughout the manuscript. First, it is incorrect that we do not show any data on pattern formation dynamics:

Supplementary Figure 4 (as well as the corresponding text on page 4 of the main manuscript) clearly indicates that a newborn African elephant does not exhibit a visible cracking pattern on its skin while it already exhibits the lattice of skin millimetric elevations (papillae). As explained in a full paragraph of the Supplementary Materials (page 17): “*This striking observation is fully consistent with the cracking hypothesis. Indeed, the formation of cracks in the stratum corneum requires that (i) the outermost part of that skin layer is sufficiently dry to allow for brittle fracture at a low critical strain, and (ii) a substantial thickening of the skin layer has occurred, allowing considerable stresses to develop. It is clear that the first condition is not satisfied while the animal is inside its mother's womb and its stratum corneum is probably fully hydrated. Moreover, it is likely that little stratum corneum thickening occurs before birth, further reducing the likelihood that the conditions for cracking are met prenatally. Thus, the absence of cracks on the skin of newborn individuals is to be expected and is confirmed by our observations, lending further support to the proposed patterning mechanism (cracking)*”.

Second, the potential effect of movement on the process of cracking is very difficult to evaluate and is unlikely to be substantial. Indeed, if cracking was due to the canonical tensile cracking (that our analysis seems to rule out), then movements might have had a considerable effect on the process. On the other hand, bending stress, as evidenced in our simulations, is very local and unlikely to be as much influenced by movements of the animal. Still, we acknowledge at the top of page 12 of the Supplementary Materials that “*spatial heterogeneities in trough spacing, skin movement or stratum corneum physical properties, may likewise contribute to different degrees of clumping in different body areas.*”

2. Theoretical aspect

From theoretical point of view, the paper is less interesting because it deals with crack formation of preexisting meshwork of valley and not the initial phase of valley itself. The phenomenon is later modification of preexisting pattern and much easier problem from pattern formation point of view. Comparison of two mechanisms should be understood by more analytical way. We can consider a elastic body whose shape is single valley (half pipe) and obtain tension at the surface by these two models analytically with given boundary condition.

Answer: We focus on how cracking of the *stratum corneum* occurs on an intricately curved substrate, *i.e.*, the pattern of millimetric skin elevations. Hence, investigating the process of papillae development is clearly beyond the scope of our manuscript. Note however that we demonstrate that papillae are formed during embryonic development because we show that newborn individuals already exhibit these structures (see Supplementary Figure 4). Note also that investigation of the mechanism generating papillae would not only be extremely difficult (if not impossible, because it would require access to living embryos of African elephants), it is also much less likely to generate original results because papillae are likely to emerge from either folding or reaction-diffusion, two processes already extensively investigated in the literature. We anyway now explicitly indicate at the end of the revised manuscript that “*... the process by which the papillae are patterned in the embryo is unknown.*”

Kondo, S., & Asai, R. (1995). A reaction-diffusion wave on the skin of the marine angelfish *Pomacanthus*. *Nature*, 376(6543), 765–768.

Rev#2

The paper by Martins et al is an elegant and exceptional paper, which it would do credit to Nature Communications to publish. It is highly detailed and innovative, and the supplemental model is, itself, more detailed, accurate and rigorously tested than many fracture models that I have seen published – it would make a quite respectable publication in its own rights. I would recommend publication of this, without any need for further modifications.

Answer: We thank very much the reviewer for her/his very positive assessment of our study.

In a couple places, I found the language a little hard to follow (particularly given my physical sciences background), but this was usually something that could be understood by a careful second reading. Nonetheless, the authors may wish to better define what they mean by trough on its first use in the second paragraph of the results, as I first understood it to mean the trough in a fold on the skin, rather than between the smaller features.

Answer: We thank the reviewers for this suggestion. We now better define the term ‘trough’ in the first paragraph of the results section. We also indicate in that paragraph (and modified Fig. 1e accordingly), that the channels in the *stratum corneum* follow troughs in the underlying network of papillae.

Second, although it is defined in the supplemental info, it would be helpful to explain the way angles are used and defined in Fig. 5 in the figure caption (i.e. that $\theta_1 > \theta_2 > \theta_3$, etc.).

Answer: We thank the reviewer to have brought this issue to our attention. As the Figure 5 caption is already large, we added a summary of the angular scatter plot approach in the Methods section of the revised main manuscript. Furthermore, we now refer to the method section in the caption of Fig. 5e.

Finally, I’m afraid that I don’t fully understand what is being presented in Fig. 4 a/c, and don’t see what features in those correspond to the cracks, or what the dashed lines by the P’s mean. All these are exceptionally minor details.

Answer: We improved clarity in Figure 4 by adding insets in Fig. 4a and 4c showing better the anisotropy of the tension field (causing cracks to occur along troughs in 4a-b and across troughs in 4c-d) and by better indicating the localisations of papillae and troughs in Fig. 4b and 4d.

As a caveat, my background is in the physical sciences, in fracture mechanics and material responses to strains. I cannot comment on the biological rigour of this paper in a professional way. I would strongly recommend that someone from a more relevant background be taken on as a reviewer as well.

Rev#3

In this manuscript the authors describe the morphological and physical structure of the african elephant's epidermis and a specific cracking behavior of it's outermost layer the stratum corneum that is thought to allow retention of mud and water. The authors have developed a potentially very interesting computational modeling tool that recapitulates the structure and cracking behavior of the stratum corneum and suggest that the specific cracking pattern is a result of orthokeratosis and reduced desquamation in combination with the specific topology of the underlying dermis resulting in local bending stress.

Answer: We thank very much the reviewer for her/his positive assessment of our study.

The description of the biophysical process as well as the model are of potentially high interest for epidermal biology not only for elephants but even for human diseased conditions. However, the causality of the biophysical features like e.g. topology of the dermal papillae, hyperkeratosis, desquamation and the cracking behavior is not completely evident, even though highly suggestive. Whereas the outcome of the computational modeling clearly supports the authors hypothesis, it is not always clear whether the assumptions match the biology.

Thus, testing of the model based on another maybe contrary biological basis like e.g. Asian elephant skin would more strongly support the hypotheses stated here.

The authors suggest that crack formation is a result of dermal curvature, orthokeratosis due to deficient desquamation and bending stress. If one of those features is really causal, the prediction is that Asian elephant would lack any of those features.

The similarities between elephant and ichthyosis patient skin that have been mentioned do suggest a cracking stimulus that is not solely based on dermal topology but e.g. the moist state of the SC as discussed in the manuscript. The authors state that based on their modeling, cracks in the elephant skin cannot be explained by desiccation. Not being an expert for computational modeling nor physic it is difficult to review whether the parameters that were changed would reproduce natural phenomena as seen in e.g. dry skin phenotypes. Thus a model based on a CT from an Asian elephant would help showing that an altered topology or altered proliferation/desquamation behavior or altered corneodesmosomal adhesion explains the lack of cracks.

This questions have in part being addressed in the supplementary section by discussing differences of forehead skin or young animals. However, those questions should be addressed at a more prominent part of the paper as they address the biological nature of the elephant skin phenotype. Moreover, a comparison with Asian elephants would show better that the model could be used to unravel variables that determine structural features of the SC or vice versa, to find causes for structural abnormalities as found in human patients.

Answer: These remarks are well taken. We unfortunately could not obtain skin samples from Asian elephants but we now include, at the end of the revised main manuscript (before the Methods section), a remark indicating that *“Further studies are also warranted to assess if the lack of a cracking pattern on the skin of the Asian elephant is due to reduced orthokeratosis (in comparison to the African elephant), preventing the stratum corneum either from drying sufficiently to break in a brittle fashion or from reaching large enough thickness to generate material failure bending stresses in the troughs.”*

Note also that we do not exactly claim that the “cracks in the elephant skin cannot be explained by desiccation”. We claim that the cracks are unlikely the result of tensile cracking (but rather of bending craking). Importantly, both mechanisms require the *stratum corneum* to be sufficiently dry to fail in a brittle fashion. This is explained in the second paragraph (starting with *“Concretely, we suggest that crack formation results from the following chain of events”*) on page 5 of the main manuscript. Please, pay particular attention to the following sentences: *“In particular, one expects strong bending stresses to develop in the troughs. As partially dry stratum corneum fails in a brittle fashion(ref9), its outermost sheets are expected to reach the material's failure value, leading to the formation of cracks”*. The remark we added at the end of the main manuscript (before the Methods section) also further clarifies this point.

Therefore, even though outside the scope of this manuscript, an analysis of human ichthyosis patients might reveal cracks being specific for desiccation, like “cracks that tend to propagate orthogonal to the direction of throughs” as shown in Figure 4. So could an analysis of SC crack behavior in human patients be used as diagnostic tool to find the origin of a barrier defect? If understood correctly the model does allow to discriminate cracking behaviors of e.g. differentially hydrated SC as suggested for different layers of the SC.

This study could be very helpful in understanding overall stratum corneum formation and function, if the different aspects of its physics, that have actually been nicely discussed the supplementary section, would be addressed more experimentally to aid the biological significance of the model.

Answer: These are fascinating suggestions but they are indeed outside the scope of our manuscript. Our anecdotal observations of images from ichthyosis and other keratinization disorder patients suggest that the cracking pattern on the skin of these patients resemble that of hierarchical tensile cracking. This makes sense as the human skin is flat, *i.e.*, it is not endowed with a network of papillae.

Minor points:

It is not clear why the authors state that African elephants show signs of orthokeratosis, being an abnormal epidermal phenotype. If there is no other elephants epidermis this is compared to, it is not clear why this is not normal african elephant epidermis with a thick stratum corneum. This may indicate an altered balance of proliferation and shedding compared to human epidermis resembling a diseased human but not elephant phenotype. Thus the wording of this paragraph should be revised.

Answer: We fully agree with the reviewer that this point warranted clarification. We now add, right after the first use of the term ‘orthokeratosis’, a sentence indicating the following:
“Note that we borrow the term ‘orthokeratosis’ from the medical literature where it refers to a series of human pathological skin conditions, whereas the thickening of the stratum corneum in African elephants is the natural physiological phenotype”.

Ichthyosis vulgaris patients are characterized e.g. by reduced fillagrin levels resulting in a brittle SC. Are fillagrin levels reduced in African elephants or is fillagrin at all expressed?

Answer: We unfortunately cannot perform quantitative assessment of filaggrin expression because of the lack of adequate samples for qPCR experiments (we tried and failed to extract RNA). We also added a sentence at the end of the revised manuscript indicating:
“However, this potential equivalence needs to be validated by detailed molecular cell biology comparisons, including the confirmation in African elephants of two key markers of ichthyosis vulgaris in humans(ref.6): the lack of keratohyalin granules in the lower stratum corneum and a substantial decrease in the expression of profilaggrin, a precursor of the protein filaggrin involved in the aggregation of keratin filaments.”

Reviewer #4 (Remarks to the Author):

In this elegant and well written study, Martins et al. present a physics lattice model to demonstrate that micron-size channels in the skin of the African bush elephant are fractures of the stratum corneum caused by local bending mechanical stresses in troughs. This finding is intriguing because, after the corresponding author's earlier study on crocodilian head skin, this report provides another example—and the first to my knowledge in mammalian epidermis—of a mechanically encoded morphology based on cracking, rather than one based on purely genetically encoded parameters (though, of course, genetic parameters may still be in part at play). This study, with quantitative imaging of a non-traditional animal model and a rigorous physical model, will be of interest to those working in a variety of fields, including the general readership of Nature Communications.

Answer: We thank very much the reviewer for her/his positive assessment of our study.

The manuscript would be strengthened by consideration of following points:

Major comments:

-The link between African elephant skin and human ichthyosis vulgaris is weak and hand-waving at best and should be scaled back. The authors should consider other ways to motivate their hypotheses.

Answer: We now emphasise on page 5 the heuristic nature of the comparison of the African elephant stratum corneum with the ichthyosis vulgaris condition in human. In addition, we now add, right after the first use of the term 'orthokeratosis', a sentence indicating the following: "Note that we borrow the term 'orthokeratosis' from the medical literature where it refers to a series of human pathological skin conditions, whereas the thickening of the stratum corneum in African elephants is the natural physiological phenotype".

-More so than the images of cracked and uncracked elephant skin, the real contribution of the manuscript seems to be in the development and implementation of the physical model. I encourage the authors to include more information about the model in the main text. Currently discussion of the model encompasses just over one page (of five) of the main text. In the manuscript's current form, the supplementary materials about numerical model could stand on their own as a full article in a more specialized journal. For publication in Nature Communications, I would like the authors to strike a better balance that allows more of their contribution (the model) to appear in the main text, to counter some of the more hand-waving arguments about comparisons to, e.g., human ichthyosis vulgaris and Giant's Causeway. The final two paragraphs of the Methods section are most intriguing to me, and I would like to see those ideas fleshed out more in the main text. Along those lines I would encourage some more of the supplementary panels to moved into parts of the main figures.

Answer: We followed the advice of the Reviewer and now included in the main text as much information about the numerical model as we could without substantially exceeding the space restrictions imposed by the journal.

- I am not convinced that the African elephant's epidermis lacks keratohyalin granules based on the data presented. Can the authors provide higher-resolution electron microscopy images and/or clearer stainings to show absence of these granules?

Answer: We adopt the most frequently used marker of keratohyalin granules (hematoxylin staining) to make our suggestion. We however agree with the reviewer that the potential correspondence between the pathological condition in human and the natural physiological phenotype of African elephants would require much additional investigation. We therefore added the following sentence at the end of the revised main text: "However, this potential equivalence needs to be validated by detailed molecular cell biology

comparisons, including the confirmation in African elephants of two key markers of ichthyosis vulgaris in humans(ref.6): the lack of keratohyalin granules in the lower stratum corneum and a substantial decrease in the expression of profilaggrin, a precursor of the protein filaggrin involved in the aggregation of keratin filaments”.

- I am not convinced that the African elephant's stratum corneum exhibits orthokeratosis, as I see small thin spots in Figure 3a,c that could be nuclei. Can the authors verify in another way that the stratum corneum is indeed anuclear?

Answer: Dark spots in the *stratum corneum* are not viable nuclei but clusters of melanin granules as can be seen in the picture below. We thank the reviewer for this remark and we now illustrate that point in the revised manuscript with a new inset of Fig. 3c (and we modified the figure legend accordingly).

Minor comments:

- Figure 4a,c needs a color bar showing what the different shades of red mean (as provided in Figure S9).

Answer: We added a color bar in Fig. 4a,c and clarified the caption accordingly.

- Can the authors explain why the elephant and simulation data in Figure 5c peak on opposite sides of 120°?

Answer: As their 95% confidence intervals overlap, our interpretation is that the differences of shape between the two probability density functions correspond to non-significant statistical variation.

REVIEWERS' COMMENTS:

Reviewer #3 (Remarks to the Author):

As already pointed out in my comments to the initial manuscript, which were consistent with reviewer 1 and 4, this manuscript requires some experimental validation/more rigorous testing of the computational model. However, the authors have basically only adjusted the text to scale down the statements and conclusions to better match the data. They unfortunately did not add any new experiments that further validate their models and address the above points that were raised by me or by these other 2 reviewers. Although the overall aim of this manuscript is interesting, in its present state the manuscript reaches insufficient significance and does in my opinion not match the requirements for a publication nature communications.

Reviewer #4 (Remarks to the Author):

I am pleased with the revisions, to both my comments and those from the other reviews.

Since the authors scaled back their emphasis on orthokeratosis to make it more suggestive rather than definitive, I request the authors to remove or soften the word in the title.